# A Web Screening on Training Initiatives in Cancer Genomics for Healthcare Professionals

**DOI:** 10.3390/genes13030430

**Published:** 2022-02-26

**Authors:** Ilda Hoxhaj, Flavia Beccia, Giovanna Elisa Calabrò, Stefania Boccia

**Affiliations:** 1Department of Woman and Child Health and Public Health—Public Health Area, Fondazione Policlinico Universitario A. Gemelli IRCCS, 00168 Rome, Italy; ilda.hoxha31@gmail.com (I.H.); stefania.boccia@unicatt.it (S.B.); 2Section of Hygiene, University Department of Life Sciences and Public Health, Università Cattolica del Sacro Cuore, 00168 Rome, Italy; flavia.beccia@gmail.com

**Keywords:** cancer genomics, training initiatives, education, healthcare professionals, literacy, web screening

## Abstract

The disruptive advances in genomics contributed to achieve higher levels of precision in the diagnosis and treatment of cancer. This scientific advance entails the need for greater literacy for all healthcare professionals. Our study summarizes the training initiatives conducted worldwide in cancer genomics field for healthcare professionals. We conducted a web search of the training initiatives aimed at improving healthcare professionals’ literacy in cancer genomics undertaken worldwide by using two search engines (Google and Bing) in English language and conducted from 2003 to 2021. A total of 85,649 initiatives were identified. After the screening process, 36 items were included. The majority of training programs were organized in the United States (47%) and in the United Kingdom (28%). Most of the initiatives were conducted in the last five years (83%) by universities (30%) and as web-based modalities (80%). In front of the technological advances in genomics, education in cancer genomics remains fundamental. Our results may contribute to provide an update on the development of educational programs to build a skilled and appropriately trained genomics health workforce in the future.

## 1. Introduction

Cancer is the leading cause of death worldwide and a major public health issue [1]. The rampant discovery of more and more genes associated with disease risk and the rapid advancement of genomic technologies led to the wide application of genomics into clinical practice [2]. Genomics enhance the understanding of human diseases, accelerate diagnosis for patients and provide increasing opportunities to tailor prevention and treatment [3]. In clinical practice, genomics has been widely used to guide the management of cancer patients by choosing a specific treatment approach based on the genetic drivers detected in the tumour rather than its histologic classification, decreasing chemotherapy toxicity and failure [4].

Nevertheless, the rapid expansion of genomics research still engenders doubt, skepticism, and challenges for healthcare professionals to fully understand and apply genomics into everyday practice. As such, knowledge translation of genomics into oncology care is a slow, thoughtful, and complex process. The increased accessibility of genomics technology is changing the educational requirements of healthcare professionals [5]. However, there is still non-conclusive evidence whether the education and training programs are adequate for the implementation of genomics medicine [6]. It was reported that healthcare professionals’ knowledge, beyond the current and projected workforce of genetic counsellors and medical geneticists, are not adequate to meet the growing demand for genomics services [6], especially in the oncology field. A recent systematic review on physicians’ knowledge in clinical cancer genomics reported limited levels of genomics literacy, which varied by specialty, type of genomics services, and years of practice [7]. If physicians do not understand the nature of genomics and how it applies to clinical practice, they may not believe the impact of such advancements in clinical practice [7]. Therefore, an appropriate and effective implementation of cancer genomics requires an adequate genomics literacy of healthcare professionals, as well as implementation of well-defined genomics core competencies [8]. In this regard, several scientific societies, universities, or research centres designed and developed initiatives to train and educate healthcare professionals in this topic. In particular, cancer genomics education is a focus of a recent European project, entitled the Innovative Partnership for Action against Cancer (iPAAC) Joint Action (JA). This JA aims to develop innovative approaches to advances in cancer control, addressing cancer prevention, approaches to the use of genomics in cancer control and care, cancer information and registries, improvements and challenges in cancer care, innovative cancer treatments, and the governance of integrated cancer control [9]. Within this JA, we provided a comprehensive map of training initiatives conducted in cancer genomics field and directed to healthcare professionals at a global level.

## 2. Materials and Methods

### 2.1. Search Strategy

We conducted a web search of online and in-person training initiatives carried out worldwide and aimed at educating healthcare professionals in the field of cancer genomics. The search was conducted using the three most used web search engines: Google, Bing, and DuckDuckGo [10]. The search was limited to initiatives publicly available in English language, from January 2003 to October 2021. This time limit was decided with the aim at identifying any possible initiative conducted after the completion of the Human Genome Project in 2003, as that was the starting point of a new era in scientific research and medical practice omics sciences related.

We used the following terms for the web search in Google using its “advanced search” application: “cancer genomics” AND “education initiatives” AND “training” AND “course” AND “healthcare professionals”. This search strategy was also used as the template for the search in Bing.

In the three search engines, after the launch of the search query, only the results in the category “All” were considered and no additional limits or filters were applied.

Two researchers (I.H. and F.B.) independently screened the identified records by title, description, and summary, whenever available, in order to identify the eligible initiatives. A database of relevant records from the screening stage was created using an Excel spreadsheet, and full texts or full web pages of these records were further assessed according to our inclusion criteria by two researchers (I.H. and F.B.) independently. Additionally, we performed a manually search on the initiatives that were suggested or mentioned in these webpages. The initiatives that satisfied the eligibility criteria were selected for inclusion in this article. Any discrepancy was solved by discussion or by the involvement of a third researcher (G.E.C.).

### 2.2. Eligibility Criteria

The eligibility criteria were formulated according to the PI/ECOS framework [11,12]:P—population: healthcare professionals;I/E—intervention/exposure: training or educational initiatives in cancer genomics;C—comparator: not applicable;O—outcome: availability of education or training initiatives.

Education or training initiatives on cancer genomics for healthcare professionals that reported information on the topic, objective, content, and target were considered eligible for inclusion. Initiatives addressing university or master students and degree courses promoted by universities were excluded.

### 2.3. Data Extraction and Synthesis of Results

For each included initiative, two researchers (I.H., F.B.) independently extracted the following data: name of initiative/project, year, country, organizer, topic, target, type of initiative, objective, and number of modules, if available. Any discrepancies in data extraction were solved by discussion or with the involvement of a third researcher (G.E.C.). The results were grouped in two categories: European and non-European initiatives. For each category, the results were summarized through a narrative descriptive synthesis [13], based on the available data extracted.

## 3. Results

The search query yielded 32,000 records on Google, 53,500 on Bing, and 149 on DuckDuckGo. Google allowed the screening of 11 pages and 100 items per page, in Bing we were able to screen 38 web pages with 10 items per page, whereas in DuckDuckGo, it was possible to explore 149 items showed in one webpage. Therefore, 1629 items were screened by title and description, of which we carefully evaluated in detail the content of 390 items. A total of 36 initiatives satisfied the eligibility criteria and were included in this work (Figure 1) [14,15,16,17,18,19,20,21,22,23,24,25,26,27,28,29,30,31,32,33,34,35,36,37,38,39,40,41,42,43,44,45,46,47,48,49]. The courses were available for all healthcare professionals willing to participate. Fifteen initiatives (39%) were conducted in European countries [14,15,16,17,18,19,20,21,22,23,24,25,26,27,28]. The majority of courses or training programs, at global level, were organized in the United States (USA) (47%) and the United Kingdom (UK) (28%). Most of the initiatives (80%) were web-based, including webinars, video lessons, online courses or workshops, and online learning material [14,17,18,19,20,21,22,23,24,25,26,27,29,30,31,33,35,36,37,38,39,40,41,42,43,44,45,46,47,48]. Two initiatives started as in-attendance course, but then were presented as online-based initiatives in the years 2020–2021 as consequence of the ongoing COVID-19 pandemic [33,34]. The first identified initiative was conducted in 2003 [29], whereas the majority of the courses (83%) were organized in the last five years [17,18,19,20,21,22,23,24,25,26,27,28,32,33,34,35,36,37,38,39,40,41,42,43,44,45,46,47,48,49]. Many of the initiatives (39%) were organized by universities [15,20,21,22,23,26,33,35,36,41,43] or professional organizations [25,28,38].

### 3.1. European Initiatives

In Europe, we identified 15 initiatives, which were conducted in the UK, Austria, Switzerland, Italy, and Germany (Table 1). Twelve initiatives were conducted online, whereas three were in-person attendance. The UK was the country with the largest number of the training initiatives (67%) on cancer genomics. Starting from 2014, the Genomic Education Program, organized by Health Education England, aimed to prepare the National Health Service (NHS) workforce to deliver the new England-wide NHS Genomic Medicine Service and to support the completion of the landmark 100,000 Genomes Project by providing annual online education opportunities in genomics, with a specific focus on genomics applications in cancer care in the module “Tumor assessment in the genomic era” [14,50,51,52,53,54]. In 2014, the European Bioinformatics Institute organized a workshop for post-doctoral researchers to introduce cancer genomics, analytical methods for detection of genome rearrangements, network analysis, and RNA-sequencing data analysis [15]. From 2014 to 2020, the Golden Helix Foundation organized several Summer Schools on genomics medicine and medical informatics for biomedical scientists and healthcare providers, also addressing cancer genomics [16,55,56,57,58]. The Guy’s and St Thomas’ NHS Foundation Trust, since 2019 and for three consecutive years, conducted online courses for healthcare professionals working in cancer care on the cancer genetic counselling, approaches to genetic testing, and the management of hereditary cancers [17,59,60].

Starting from the collaboration with Guy’s and St Thomas’ NHS Foundation Trust, the Medics.Academy developed a four-hour online courses available in Europe on cancer genomics and precision oncology [18].

The EMBL-EBI Training, which is part of the ELIXIR infrastructure, ran a Cancer Genomics course in May 2021. The course was made available online to be accessible any time. It is composed of pre-recorded lectures, presentations, and practical sessions to learn about cancer genomics and enhance skills in the analysis of cancer genomics data [19].

The NHS Health Education England launched a Massive Open Online Course (MOOC) in 2021 titled “Whole Genome Sequencing—Decoding the Language of Life and Health” on genome sequencing, DNA replication, next-generation sequencing, risk assessment, and data sharing directed at healthcare professionals [14]. Other MOOCs were organized in 2020–2021 by several universities on cancer genetics and pharmacogenetics [20]; epigenetics and cancer diagnosis [21]; genomics data and ethical, legal, and social related-issues; genomics in clinical practice; counselling; next-genomics sequencing; and genome-wide association studies [22,61,62,63]. The Royal College of General Practitioners organized webinars regarding the impact of genomics in primary care and basic concepts on familial cancer, rare diseases, antenatal care, testing, and ethical issues [23].

In Austria, within the Corbel Project, a series of webinars were conducted on cancer genomics research and clinical data for data managers, researchers, and postdocs involved in clinical, translational, and biomedical research [24,64]. Precision Medicine was the focus of the MOOC organized in 2017 in Germany for researchers, policy administrators, and life science professionals [25] and of the MOOC conducted in 2021 in Switzerland, for primary care physicians, cancer and non-communicable diseases specialists, public health policy and decision makers, biomedical researchers, and drug developers [26]. Recently in Italy, a multi-disciplinary Delphi method of experts in the field defined the core competencies on cancer genomics for physicians and nurses, which were grouped into three categories: knowledge, attitudes, and practical abilities [8]. The defined competencies were then used to elaborate a distance-learning course in English language at the European level, entitled “Oncogenomics for health professionals”. This online course was aimed at improving the knowledge, attitude, and practice of physicians on the fundamental principles of genetics and on the major clinical applications of genomics technologies in oncology, based on the most recent scientific evidence [27].

In addition to these initiatives at national level, the European Society of Medical Oncology organized in 2019 a workshop for its members regarding hereditary cancer syndromes, susceptibility to develop cancer, and clinical management [28,65].

**Table 1 genes-13-00430-t001:** Training initiatives on cancer genomics for healthcare professionals conducted at the European level.

Training Initiative/Program [Ref]	Country	Year	Organizer	Courses/Modules [Ref]	Topics	Target	Type	Status (at 20 January 2022)
Genomics Education Programme [14]	UK	2014–ongoing	Health Education England	(1) Genomics 101: Genomics in Healthcare [50] (2) Genomics 101: Investigating the Genome Part 1: The Process [51] (3) Whole Genome Sequencing: Decoding the Language of Life and Health [52] (4) Tumour Assessment in the Genomic Era [53] (5) Intermediate Genomics Access Course [54]	Knowledge, skills, and experience on genomics and cancer genomics	NHS workforce	Online courses	Open
The European Bioinformatics Institute (EMBL-EBI) Workshop [15]	UK	2014	European Bioinformatics Institute (EMBL-EBI)—Wellcome Genome Campus	Cancer Genomics Workshop [15]	Data analysis of cancer genomics data	Ph.D. students and post-doctoral researchers	In attendance/workshop	Closed
The Golden Helix Summer Schools [16]	UK	2014–2020	Golden Helix Foundation	(1) 2020 Golden Helix Summer School [58] (2) 2018 Golden Helix Summer School [55] (3) 2016 Golden Helix Summer School [56] (4) 2014 Golden Helix Summer School [57]	Genomic Medicine, Cancer genomics, Biomedical informatics	Biomedical scientists and healthcare providers	Summer School	Closed
The Future of Genomics and Precision Medicine [25]	Germany	2017	ASCO Leadership Team	The Future of Genomics and Precision Medicine [25]	Precision medicine, Cancer genomics, and Big data	Care team members, health and policy administrators, researchers, and life sciences professionals	MOOC	Closed
Royal College of General Practitioners GENOMICS WEBINAR [23]	UK	2018	Royal College of General Practitioners	(1) Part 1: Familial cancer (2) Part 2: Rare disease (3) Part 3: Non-Invasive Prenatal Testing (4) Part 4: Ethical issues	Genomics and clinical scenarios (Familial Cancer, Rare Disease, Antenatal care, commercial testing, and ethical issues)	General Practitioners	Webinar series	Closed
CORBEL webinar series [24]	Austria	2019	CORBEL project (ELIXIR & BBMRI)	Genomics and clinical data at your fingertips with open-source software: tranSMART & cBioPortal [64]	Cancer research, genomics, and clinical data	Data managers, researchers, Ph.D. students and postdocs involved in clinical, translational, and biomedical research	Webinar	Closed
ESMO Meeting Preceptorship [28]	EU LEVEL (hosted in Switzerland)	2019	European Society of Medical Oncology (ESMO)	Hereditary Cancer Genetics [65]	Hereditary cancer syndromes, susceptibility to develop cancer, clinical management	ESMO members (oncologists)	In attendance/workshop	Closed
Guy’s and St Thomas’ NHS Foundation Trust [17]	UK	(1) 2019 (2) 2021	Guy’s and St Thomas’ NHS Foundation Trust (UK)	(1) Cancer Genetics Course [59] (2) Virtual Cancer Genetics Course [60]	Cancer genomics and cancer genetic counselling, genetic testing, management of hereditary cancers, and consent taking	Healthcare professionals working in genetics or primary care/specialist settings such as oncology, breast care, gynae-oncology, gastroenterology, and screening services	Online course	Closed
Cancer in the 21st Century—the Genomic Revolution [21]	UK	2020	University of Glasgow	Cancer in the 21st Century—the Genomic Revolution [21]	Cancer diagnosis and treatment, epigenetics	Healthcare professional, students, nurses, physicians	MOOC	Open
Using Personalised Medicine and Pharmacogenetics [20]	UK	2020	University of East Anglia	Using Personalised Medicine and Pharmacogenetics [20]	Cancer genetics, pharmacogenetics	Qualified clinicians (general practitioners, oncologists, general physicians, pharmacists, nurse practitioners, clinical scientists) and scientists (biologists and bioinformaticians)	MOOC	Open
Genomics in Healthcare Program [22]	UK	2021	St George’s University of London	(1) The Genomics Era—the Future of Genetics in Medicine [61] (2) Genomic Technologies in Clinical Diagnostics: Molecular Techniques [62] (3) Genomic Technologies in Clinical Diagnostics: Next Generation Sequencing [63]	(1) Human genome, genomic data, genomics in clinical practice, genetic counseling, ELSI of genomics, cancer genomics (2) Molecular genetic techniques and their application, single nucleotide polymorphism (SNP) genotyping and genome wide association studies (GWAS) (3) Next-genomics sequencing	Scientists and healthcare professionals at a postgraduate level	MOOC	Open
Precision Medicine [26]	Switzerland	2021	University of Geneva	Precision Medicine Course [26]	Genetic counseling, cancer, monogenic and complex diseases, risk assessment, pharmacogenomics, multi-omics data integration	Primary care physicians and other first-line healthcare professionals, cancer and non-communicable disease specialists, public health policy and decision makers, biomedical researchers, and drug developers	MOOC	Open
Oncogenomics for health professionals [27]	Italy	2021	Italian National Institute of Health	Oncogenomics for health professionals [27]	Cancer genomics	Physicians, biologists	Online course	Closed
Cancer genomics [19]	UK	2021	EMBL—EBI Training (in ELIXIR Training Platform)	Cancer genomics [19]	Cancer genomics and data	Healthcare professionals	Online course	Open
Cancer Genomics: The essentials [18]	UK	2021	Medics.Academy	Cancer Genomics: The Essentials [18]	Cancer biology and genomics, cancer genomics, genomic testing, precision oncology	Consultants, general practitioners, doctors, nurses, allied healthcare professionals	Online course	Open

### 3.2. Non-European Initiatives

We identified 21 non-European initiatives [29,30,31,32,33,34,35,36,37,38,39,40,41,42,43,44,45,46,47,48,49], 17 conducted in the USA [29,30,31,32,33,34,35,36,37,38,39,40,41,42,43,44,45], 2 at a global level [46,47], 1 in Africa [48], and the other 1 in Australia [49], reported in Table 2. Sixteen initiatives were organized online: two were in-person attendance and three adopted a mixed approach. At the global level, the Global Genomic Medicine Collaborative (G2MC) created in 2020 the Education Working Group, that aim to develop courses, assessment tools, and resources on the implementation of genomics medicine in clinical practice and the integration of genetic counselling in oncology care [46,66]. Since 2018, the Global Genomics Nursing Alliance (G2NA) organizes quarterly webinars on genetic counselling, genomics in clinical practice, cancer genomics in nursing care, and capacity building to the members [47].

In the USA, the Cancer Genomic Education Program, through different annual training modalities, aimed to equip primary care physicians and other healthcare professionals with the necessary knowledge and skills to help patients understand their personal cancer risk [42,67,68,69]. The National Cancer Institute (NCI) and The National Institutes of Health (NIH) organized the Translational Research in Oncology Course from 2003 to 2019, designed to provide an overview of cancer biology, treatment, epidemiology, mechanisms, and the identification of novel molecular targets [29,70]. Additionally, the NIH created the Cancer Information Summaries on a wide range of cancer topics for oncologist and clinicians and, in 2013, funded the IGNITE (Implementing GeNomics In pracTicE) Network to support the development, investigation, and dissemination of the genomic medicine practice model. Within the project, the IGNITE Toolbox was created as an open source tool of peer-reviewed genomics medicine implementation. For clinicians, this tool provides background information, benefits of adopting genomics medicine in patient care, and key challenges and stakeholders to consider. For researchers, it provides sample consent forms, surveys, data dictionaries, and other resources to help the implementation of science research in genomics medicine [39,40,71,72,73,74,75,76,77,78,79,80].

In 2010, the Training Residents in Genomics (TRIG) Working Group was formed, aiming to develop teaching tools and promote genomics education to pathology residents. Supported by grants from the NCI, genomic workshops and courses, online modules, and the Universal Genomics Instructor Handbook and Toolkit were provided to professionals of several specialties as well, such as neurology, ophthalmology, oncology, and cardiology [32,81,82,83].

In 2013–2014, the National Human Genome Research Institute (NHGRI) organized a series of lectures about genomics in medicine, focusing also on the application of genomics in cancer care and prevention, and created a free online repository called the “Genetics/Genomics Competency Center”, which consists of more than 500 genomics educational materials that are continuous updated [31,84,85,86,87,88,89,90,91,92].

The American Society of Clinical Oncology (ASCO) created the Genetics Toolkit in 2015 with the objective to provide oncologists with the necessary tools and resources that will assist them in effectively integrating hereditary cancer risk assessment, for the patients and their relatives, into everyday practice [38,93,94].

Empowering oncologists and clinicians to integrate genomics into their clinical practice was the objective also of the Maine Cancer Genomics Initiative. This initiative, undertaken by the Jackson Laboratory, developed several online programs and resources and in-person interactive workshops in clinical genomics, testing, and risk assessment, especially for screening protocols for hereditary tumours [37,95,96,97].

After identifying a need for further awareness and training among primary care providers about appropriate referral for BRCA counseling and testing, the Michigan Cancer Genomics Program collaborated with federal, state, and local partners to launch in 2014 a free online continuing medical education course and other online tools for healthcare professionals for increasing cancer genetic literacy among the public and healthcare providers, improving the use of appropriate cancer risk assessment and clinical genetics services, enhancing communication, and developing partnerships with cancer genetic service providers and other key stakeholders [45].

In 2019, the American College of Medical Genetics and Genomics, through online courses, offered geneticists valid instruments to identify common genetic syndromes, provide genetic counselling for common human cancers, and explain clinical and molecular aspects of inherited cancer syndromes [35].

The Global Genetics and Genomics Community (G3C) created the learning portal “Genomic Health care Simulations” for practicing healthcare professionals, which is a bilingual collection of interactive cases that demonstrate how genetics and genomics link to health and illness [44].

In August 2009, the Cancer Genomics Consortium (CGC) was formed by a group of clinical cytogeneticists, molecular geneticists, and molecular pathologists interested in education and promoting best practices in clinical cancer genomics. In collaboration with the University of Wisconsin, the Consortium offered a series of webinar “Cancer Genomics Consortium 2020–2021” to genetic counsellors and healthcare professionals who generate or use genomics/genetics data in their practice. These webinars included resources to aid interpretation of cancer variants, germline predisposition to cancer, the use of new technologies in clinical genomic testing, and the detection of copy number abnormalities from NGS data in cancer samples [36]. In 2018, the Mayo Clinic, in collaboration with the University of Illinois, offered an intensive course for scientists and clinicians covering the basics of computational genomics, genome sequencing, and assembly, polymorphism, and variant analysis, epigenomics, and data visualization [34,98,99]. In the last years, other universities in the USA organized annual online courses on cancer genomics and sequencing, precision oncology, such as Stanford Genetics and Genomics Program, HMX Pro Genetics Cancer Genomics and Precision Oncology, and Short Courses of Genetic Analysis [33,41,43,100,101,102,103,104,105,106,107,108,109,110,111,112,113,114].

In Africa, in 2017, the African Genomic Medicine Training Initiative was created by volunteers globally, supported by H3ABioNet, the Southern African Human Genome Programme, and the University of Pretoria. This initiative designs and develops Genomics Medicine training for African-based healthcare professionals on genomics applications in cancer care, pharmacogenomics, counselling, and clinical genomic research [48,115].

In Australia, the Genomics basics for Primary Care was created in 2019 as a joint initiative between Metro North GPLO and the Brisbane North Primary Health Network. It aimed to increase the knowledge of general practitioners of new developments in diagnostic cancer genomics so that they can understand how to support their patients if a referral is not necessary [49].

**Table 2 genes-13-00430-t002:** Training initiatives on cancer genomics for healthcare professionals conducted at non-European level.

Initiative [Ref]	Country	Year	Organizer	Courses/Modules [Ref]	Topics	Target	Type	Status (at 20 January 2022)
NIH: Education and Training for Health Professionals [29]	USA	2012–2019	National Cancer Institute (NCI) and National Institutes of Health (NIH)	Translational Research in Oncology Course (TRACO) [70]	Cancer genomics	Healthcare providers	Videocast	Open
NHGRI: The 2013–2014 Genomics in Medicine Lecture Series [30]	USA	2013–2014	National Human Genome Research Institute (NHGRI)	(1) When the Lifeguard for the Gene Pool Goes on Strike: DNA Repair Disorders Xeroderma Pigmentosum and Trichothiodystrophy (2) Genetics and Genomics of Thyroid Neoplasms: Moving Closer Towards Personalized Patient Care (3) Practicing Precision Medicine in Cancer Using Genomics (4) Targeting the Genetic Basis of Kidney Cancer: A Metabolic Disease (5) Integration of Genomics into Nursing Practice (6) Cancer Genomics and Precision Medicine in the 21st Century (7) The New Telomere Diseases: Organ Failure and Cancer (8) Genome and Transcriptome Dynamics in Cancer Cells	Oncology and genomics	Healthcare professionals	Lecture series (in attendance)	Closed
Genetics/Genomics Competency Center (G2C2) [31]	USA	2008–2016	National Human Genome Research Institute (NHGRI)	(1) Clinical genetics resources [84] (2) Ethical, legal and social implications resources [92] (3) Family history resources [86] (4) General genomics resources [87] (5) Genetic conditions resources [88] (6) Genetic counselling resources [89] (7) Genetic and genomic testing resources [90] (8) Pharmacogenomics resources [85] (9) Risk assessment resources [91]	Genomics, clinical genetics and cancer	Healthcare educators and practitioners	Online repository of genomics educational materials such as YouTube videos, online courses	Open
Training Residents in Genomics (TRIG) Working Group [32]	USA	2016—Updated in 2020	National Institutes of Health (NIH)	(1) Genomics workshops and courses [81] (2) Online Genomic Pathology Modules [82] (3) Universal Genomics Instructor Handbook and Toolkit [83]	Genomics and cancer	Residents	Online courses	Open
African Genomic Medicine Training Initiative [48]	Africa	2017	Working Group made up of volunteer globally, supported by H3ABioNet, Southern African Human Genome Programme, University of Pretoria	Course Description: Introduction to Genomic Medicine for Nurses in Africa Training—2017 [115]	Genomics applications, pharmacogenomics, clinical research, genetic counseling in cancer and clinical care, genetics	Nurses, researchers in genomics/genetics field, experts in education/training	Online meetings, online course	Closed
Global Genomics Nursing Alliance (G2NA) Webinars [47]	GLOBAL	2018–2021	Global Genomics Nursing Alliance (G2NA)	2018–2021 G2NA Webinar series [47]	Genetic counseling, genomics in clinical practice, competence building, engagement	G2NA members (nurse)	Webinar series	Open
Short Courses of Genetic Analysis [33]	USA	2018–2021	Rockefeller University	(1) 2020 Genetic Association Course [100] (2) 2018, 2019, 2020, 2021 Advanced Gene Mapping Course [101] (3) 2018, 2019, 2021 Complex Trait Analysis of Next Generation Sequence Data [102,103,104] (4) 2018 [105]	Genomics in clinical and cancer care	Healthcare professionals	2018–2020 in attendance; 2021 online	Open
Mayo Clinic & Illinois Alliance for Technology-Based Healthcare [34]	USA	(1) 2021(2) 2018	Mayo Clinic & Illinois Alliance	(1) Introduction to Computational Genomics [98] (2) Computational Genomics Course [99]	Computational genomics, genome sequencing and assembly, polymorphism and variant analysis, cancer genomics, epigenomics, systems biology, data visualization	Scientists and clinicians	(1) Online course, (2) in attendance workshop	Closed
American College of Medical Genetics and Genomics: Genetics and Genetics and Genomics Review Course [35]	USA	2019	American College of Medical Genetics and Genomics	Genetics and Genomics Review Course [35]	Cell Biology/Genomics, Molecular Biology, Laboratory Genetics and Genomics, Genetic Transmission, Developmental Genetics, Cancer Genetics, Genetic Counseling and Risk Genomic Medicine	Genetics healthcare professionals	Online course	Closed
Genomics basics for Primary Care [49]	Australia	2019	Metro North GPLO, and the Brisbane North Primary Health Networ, supports by Queensland Genomics	Genomics basics for Primary Care: cancer genomics for GPs [49]	Genomics in primary care and cancer	General Practitioners	In attendance workshop	Closed
Global Genomic Medicine Collaborative (G2MC): Education Working Group [46]	GLOBAL	2020	Global Genomic Medicine Collaborative (G2MC)	G2MC Grand Rounds on Genomic Medicine Implementation [66]	Genomic medicine in cancer and clinical practice	Medical practitioners, young investigators	Video series	Open
Cancer Genomics Consortium 2020–2021 Webinars [36]	USA	2020–2021	Cancer Genomics Consortium and the University of Wisconsin	(1) Circulating tumor DNA testing in oncology: established and emerging approaches, clinical utility and applications (2) Resources for Interpreting Cancer Genomes: From Knowledgebases to Gene Lists (3) Copy Number Abnormality Detection from Whole Genome Sequencing Data (4) Interpretation of Copy Number Abnormalities (CNAs) and CN-LOH in Cancer using the ACMG/CGC Standards: Working through Complex Cases (5) Optical Mapping and its Role as a Cytogenomics Tool in Cancer (6) Application of Optical Mapping for Comprehensive Assessment of Structural Rearrangements in Hematological Malignancies	New technologies in clinical genetic/genomic testing, interpretation of sequence and copy number variants, germline predisposition to cancer, tumor testing	PhD, MD, RN, Genetic Counselors, and healthcare professionals who generate or use genomic/genetic data in their practice	Webinar series	Open
The Maine Cancer Genomics Initiative (MCGI) [37]	USA	2021 (updated)	Jackson Laboratory	(1) Genomic tumor boards [95] (2) Precision Oncology online courses [96] (3) Tools and factsheets [97]	Clinical genomics, tumor testing, risk assessment	MCGI oncologists, clinicians	Genomic tumor boards; online courses; in-person forums; tools and factsheets	Open
American Society of Clinical Oncology—Genetics Toolkit [38]	USA	2021	American Society of Clinical Oncology (ASCO)	(1) Genetics and Genomics Course Collection (15 sub-courses) [94] (2) Molecular Oncology Tumor Boards (10 sub-courses) [93]	Cancer genomics, risk assessment, counseling, genetic testing	Oncologists	Online courses	Open
National cancer institute: pdq^®^ Cancer Information Summaries [39]	USA	2021 (updated)	National Institutes of Health (NIH)	(1) Cancer Genetics Overview [72] (2) Cancer Genetics Risk Assessment and Counseling [75] (3) Genetics of Breast and Gynecologic Cancers [71] (4) Genetics of colorectal cancer [73] (5) Genetics of endocrine and neuroendocrine neoplasms [78] (6) Genetics of renal cell carcinoma [77](7) Genetics of prostate cancer [74] (8) Genetics of skin cancer [75]	Cancer genomics and genetics	Clinicians, oncologists	Information summaries	Open
Implementing Genomics in Practice (IGNITE) Network TOOLBOX [40]	USA	Ongoing	National Institutes of Health (NIH)	(1) Disease diagnoses, risk assessment, pharmacogenomics, for clinicians [79] (2) Data collection, laboratory testing, research tools for researchers [80]	Cancer genomics and genetics	Genetic counselor, nurse, pharmacists, physician, physician assistant	Open-source tool	Open
Genetics and Genomics Program [41]	USA	Ongoing	Stanford Genetics and Genomics	(1) Fundamentals of Genetics: The Genetics You Need To Know [106] (2) Genomics and the Other Omics: The Comprehensive Essentials [107] (3) Principles and Practices of Gene Therapy [108](4) Understanding Cancer at the Genetic Level [109] (5) Genetic Engineering and Biotechnology [110] (6) Stem Cell Therapeutics [111] (7) Personal Genomics and Your Health [112] (8) New Frontiers in Cancer Genomics [102] (9) Epigenetics and Microbiomics in Precision Health [103]	Genomics, personalized medicine, cancer genomics, regenerative medicine, epigenetics, DNA sequencing technologies, commercial applications of genetics research	Medical practitioners	Online courses	Open
Cancer Genomics Education Program [42]	USA	Annually ongoing	The City of Hope community— Division of Clinical Cancer Genomics	(1) Intensive Course in Cancer Risk Assessment [67] (2) Clinical Cancer Genomics Community of Practice [68] (3) Cancer Genomics Career Development Program [69]	Cancer molecular genetics, genetic cancer risk assessment, genetic testing process	Primary care physicians and other healthcare professionals; members of Clinical Cancer Genomics Community of Practice	Training, educational webinars, workshops	Closed
HMX Pro Genetics Cancer Genomics and Precision Oncology [43]	USA	Annually ongoing	Harvard Medical School	Cancer Genomics and Precision Oncology [43]	Cancer genomics, tumor sequencing, precision oncology	Science, business, and medical professionals	Online course	Open
Genomic Health Care Simulations [44]	USA	Ongoing	Global Genetics and Genomics Community (G3C)	Unfolding Case Studies for Genetics & Genomics Healthcare Education [44]	Genetic risk, cancer risk assessment	Practicing healthcare providers	Learning portal	Open
Michigan Cancer Gemomics Program [45]	USA	2003–ongoing	Michigan Department of Health and Human Services	Cancer Genomics Program [45]	Cancer genomics, risk assessment, clinical genetics services, counseling	Healthcare providers, primary care providers	Online courses and tools	Open

## 4. Discussion

The need to train healthcare professionals in cancer genomics is a natural consequence of the disruptive development in this field since the sequencing of the human genome. From an oncologic perspective, major results were achieved in the last decades, enabling medicine to offer cancer patients targeted treatments and innovative approaches [116]. These rapid changes require skilled healthcare professionals for the provision of the optimum care to patients. In this context, the objective of our study was to summarize the initiatives aimed at improving healthcare professionals’ literacy in cancer genomics field. Even though the aim was not to assess which countries are at the forefront in healthcare professionals training on cancer genomics, the results suggest that greater attention to this topic was paid in the USA and the UK, although the results might be influenced by the search strategy adopted. Most of the initiatives were directed to non-geneticist healthcare professionals and were offered in web-based modalities, such as online lectures, courses, resources, or tools. Online training format was considered as the more effective modality to enhance knowledge, skills, and confidence [117,118,119].

The technological innovations of genomics and other omics sciences allow physicians to offer personalized diagnosis and treatments, especially in the oncology field [120,121,122]. Through personalized medicine based on genomics, it has been possible to achieve an improvement in risk stratification, for example, in hereditary breast cancer. Carriers of BRCA1/2 pathogenic variant carriers and their relatives have an increased risk of breast cancer due to their genetic predisposition and are a well-established sub-group of individuals with needs for targeted prevention and care pathways based on a risk-based approach [123].

Furthermore, researchers are using whole exome analysis to characterize the genomic landscape of responders and non/responders to anti-HER2 treatment among metastatic breast cancer patients [124]. Genomics is contributing to the understanding of the polygenic nature of certain diseases, such as cardiovascular disease or diabetes. Polygenic score models are being developed for specific subgroups of cardiovascular disease, such as CAD, stroke, and hypertension, that are currently used for screening and risk assessment to provide inform decisions for targeted treatments or tailored lifestyle modifications. Polygenic scores might be integrated into national programs to improve the predictive accuracy of cardiovascular risk assessment at population level [125].

However, due to the shortage of genetic professionals, citizens and patients were mostly directed to non-geneticist healthcare professionals, regardless of their field of expertise, for genomics test interpretation or counselling [126]. Therefore, in this context, it is necessary to increase training efforts for all the healthcare professionals, enabling them to better support their patients with appropriate evidence-based health decisions. In particular, training initiatives should focus on general practitioners and primary care physicians, as the first contact point for patients for counselling, genomics test interpretation, or family risk assessment. Primary care practitioners expressed the necessity for education activities in cancer genomics in order to answer adequately citizens’ questions [127,128,129], especially for direct-to-consumer genetic tests that are purchased online without medical supervision or counselling [130,131,132,133,134].

Education strategies should cover the challenges posed by the increasing demand for cancer genomics competencies in clinical care regarding the use, implementation, and validity of genomic tools. Therefore, in response to these challenges, several organizations, such as the Inter-Society Coordinating Committee for Practitioner Education in Genomics (ISCC-PEG), have been created to identify educational needs and potential solutions. Regardless of the increased attention by different organizations, further training efforts are needed in the field.

Furthermore, several researchers have suggested the integration of genomics into curricula for undergraduate and postgraduate studies and their continuous update [135,136]. A survey conducted at the global level in 2018, only three countries (USA, the UK, and Japan) incorporated genomics into nursing education [137]. A systematic review included 41 articles, summarized the characteristics of genomics curricula for health professional students, reported that the majority of the curricula were offered to medical and pharmacy students and were not theory-based, and 85.4% of the studies did not report follow-up data regarding the evaluation outcomes [138]. Despite the increased interest in evaluating genomics knowledge, it remains substantial to understand the possibilities of healthcare professionals to undertake training options, given the time restrains in their healthcare practice. In order to develop effective training opportunities, the impact of such initiatives on healthcare professionals’ knowledge, abilities, attitudes, and skills should be explored. In this context, future studies should evaluate the outcomes of the training options, reporting measured indicators, that could help to improve the development of new initiatives, based on education needs.

Given that cancer genomics is a developing field, curricula should be elaborated based on basic concepts and competences required for the accurate use, interpretation, and dissemination of genomics information. Future studies should pay attention to the current structure of curricula to better understand whether the new healthcare professional figures will have the required core competencies to cope with the challenges that genomics has been posing to healthcare systems.

Notwithstanding, in front of the rapid evolving knowledge in cancer genomics, constant training and updating of such curricula and education initiatives remain fundamental, as it is shown in the Italian example [27].

This work has some limitations that should be mentioned. Using only search terms in the English language, we might have not been able to identify initiatives that were conducted at in national language by different countries. It should be noted that not all the conducted initiatives could be publicly available across the Internet or presented in organizational repositories, thus suggesting that not all of them were possible to identify. Additionally, although we performed the search in the most used search engines worldwide, we might have lost initiatives available at other search engines, such as WeChat or Baidu, mostly used in Eastern countries, suggesting that our results might not be representative of all European countries. Moreover, the high heterogeneity of the identified initiatives, in terms of methodology and topics, and the lack of detailed information in some of them, did not allow us to make an accurate comparison among them. Furthermore, it was not possible to assess the effectiveness of the initiatives, since they did not provide any quantitative data, measures, or indicators.

Despite these limitations, our work is the first attempt to summarize past and ongoing initiatives addressing healthcare professionals in the cancer genomics field using a web search with a systematic and scientific approach. This innovative web-based screening methodology was recently applied in a previous study aimed at mapping educational initiatives to increase citizens’ literacy in genomics and genetics field [139]. Focusing on both healthcare professionals and citizens’ literacy is a priority strategy for implementing PM in clinical care and improving the public understanding of its value for heath.

## 5. Conclusions

Understanding and appropriately being able to apply genomics tools in the management of cancer, in terms of prevention, diagnosis, and treatment, is the key to coping with innovation to provide quality in care. Our results may contribute to provide an update on the development of educational programs to build a skilled and appropriately trained genomics health workforce in the future.

## Figures and Tables

**Figure 1 genes-13-00430-f001:**
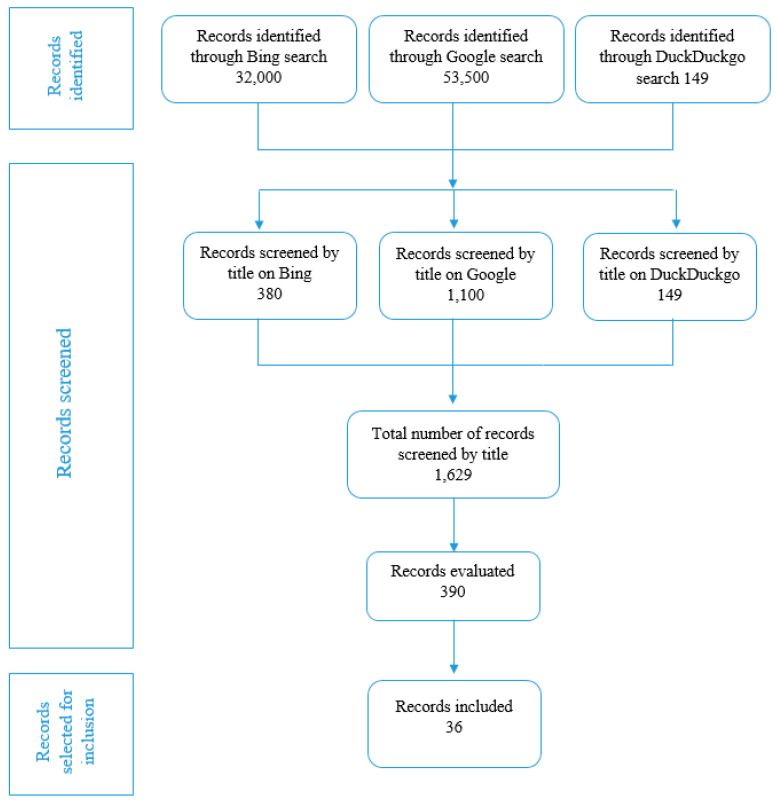
Flow chart of the search strategy.

## Data Availability

Not applicable.

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
