# Peer review of "A Web Screening on Training Initiatives in Cancer Genomics for Healthcare Professionals"

_genes, 2022, doi:10.3390/genes13030430_

Round 1

Reviewer 1 Report

The manuscript "A web screening on training initiatives in cancer genomics for healthcare professionals" by Hoxhaj and colleagues aims at capturing the current offer of resources for cancer genomics, intended for medical professionals. While the paper topic is innovational and interesting, there are some major issues that flaw its intended nature of a "original research paper". Following, my points:

Major:

- While I understand the purpose of the paper and my heart is with the spirit the authors intended (i.e. the need for more genomics in medicine, and analyzing initiatives to improve the current situation), I believe the study requires more hard data than currently reported. As the authors point out (line 322), the lack of a quantitative analysis renders the study more an opinion piece than an actual research article. The authors should investigate the number of BSc and MSc courses covering (or specifically focused on) genomics worldwide, and for example how they are distributed within the classic University courses of Biology and Medicine.

- In addition to the previous point, the piece must benefit from an unbiased statistical survey on the interest, amongst appropriate healthcare personnel, towards the topic of cancer genomics. This must assess the key points covered by the paper, broadly intended as: 1) are you aware of the importance of genomics in medicine? 2) have you sought out courses or online material to improve your knowledge on the topic? 3) do you think there is a vast offer for genomics courses online? 4) have you attended such courses?

- Concerning the web serch performed by the authors: Google and Bing are indeed very popular search engines, but they are heavily biased by design: by the geographic location, by the user's previous browsing history (the so-called "cookies" present in the web browser) and by the presence of advertised links. In order to perform a scientifically sound study, the authors must add the results from an unbiased, user-independent search engine such as DuckDuckGo.

- Lines 302-310: the authors claim that only three countries (USA, UK and Japan) incorporated genomics into student classes as of 2018. This is highly incorrect, as, alredy in 2018, there were several university classes on genomics across Europe, and even entire BSc/MSc courses named "Genomics" (in Leuven, Paris, Bologna, Dublin, Milano, Nijmegen...)

Minor:

- Line 303: typo "continously" should be "continuous"

Author Response

Dear Editor, Dear Reviewer,

We would like to thank you for your valuable comments and for the opportunity to resubmit our work. We have amended the paper according to the received suggestions and we hope that it now appears improved.

Hereafter point-to-point answers are provided.

Best regards

Reviewer 1

Comment 1: While I understand the purpose of the paper and my heart is with the spirit the authors intended (i.e. the need for more genomics in medicine, and analyzing initiatives to improve the current situation), I believe the study requires more hard data than currently reported. As the authors point out (line 322), the lack of a quantitative analysis renders the study more an opinion piece than an actual research article. The authors should investigate the number of BSc and MSc courses covering (or specifically focused on) genomics worldwide, and for example how they are distributed within the classic University courses of Biology and Medicine.

Answer: We thank the reviewer for this very valuable comment. We would like to specify that the objective of our study was to research and summarize the training initiatives conducted worldwide in the field of cancer genomics for healthcare professionals. Moreover, we conducted our research with an "innovative" method through a web search, while still applying a rigorous methodology linked to systematic reviews. This is certainly an element of originality of our work. We had already applied this methodology in our other study previously published and also cited in this paper (Sassano M, Calabrò GE, Boccia S. A Web Screening on Educational Initiatives to Increase Citizens' Literacy on Genomics and Genetics. Frontiers in Genetics; 12. Epub ahead of print July 7, 2021). Unfortunately, our work has limitations that we have reported in the discussion. Among these, the lack of a quantitative analysis is an important limitation but unfortunately as reported in the paper"... the high heterogeneity of the identified initiatives, in terms of methodology and topics, and the lack of detailed information in some of them, did not allow us to make an accurate comparison among them. Furthermore, it was not possible to assess the effectiveness of the initiatives, since they did not provide any quantitative data, measures, or indicators ". We specify, however, that the objective of the study was not to quantitatively analyze the effectiveness of the initiatives but to describe an overview of the training initiatives on cancer genomics aimed at healthcare professionals conducted worldwide. All in order to understand the current burden of training in this particular sector of public health. Furthermore, we agree with the reviewer on the importance of investigating the number of BSc and MSc courses covering (or specifically focused on) genomics worldwide, and for example how they are distributed within the classic University courses of Biology and Medicine. However, this was not the aim of our study that instead was aimed at investigating the training offer on cancer genomics for healthcare professionals. We fully agree that the level of university training should be evaluated in order to understand whether the new healthcare professional figures will be ready to cope with the challenges that genomics has been posing to healthcare system, therefore we added this suggestion at our discussion section.

Comment 2: In addition to the previous point, the piece must benefit from an unbiased statistical survey on the interest, amongst appropriate healthcare personnel, towards the topic of cancer genomics. This must assess the key points covered by the paper, broadly intended as: 1) are you aware of the importance of genomics in medicine? 2) have you sought out courses or online material to improve your knowledge on the topic? 3) do you think there is a vast offer for genomics courses online? 4) have you attended such courses?         

Answer: We thank the reviewer for this comment. We searched the literature about the current level of education among healthcare professionals and their educations needs. We summarized the findings at the introduction part, from line 2 to 15 at page 3, reporting that healthcare professionals’ knowledge is not adequate to meet the growing demand for genomics services, and also at the discussion section, indicating references 121-128.        

Regarding the suggestion of conducting a survey to evaluate the key points covered by the paper (1. are you aware of the importance of genomics in medicine? 2. have you sought out courses or online material to improve your knowledge on the topic? 3. do you think there is a vast offer for genomics courses online?), it would require additional work that we will certainly be able to evaluate for our future research activity.

However, we totally agree that it remains substantial to understand the possibilities of healthcare professionals to undertake training options, given the time restrains, and their daily healthcare practice. Moreover, it is important to evaluate whether the courses offered had an impact in improving their knowledge, abilities and skills. We added this suggestion at our discussion, indicating the need for further studies, with quantitative data and indicators that could help the development of effective training opportunities.

Comment 3: Concerning the web serch performed by the authors: Google and Bing are indeed very popular search engines, but they are heavily biased by design: by the geographic location, by the user's previous browsing history (the so-called "cookies" present in the web browser) and by the presence of advertised links. In order to perform a scientifically sound study, the authors must add the results from an unbiased, user-independent search engine such as DuckDuckGo.

Answer: We thank the reviewer for the suggestions. We conducted the search in Google and Bing being the most popular search engines worldwide. The search string used in these website, using advances search is as follows:  
a) in Google:  https://www.google.it/search?hl=en&as_q=cancer+genomics+education+initiatives+training+and+course+for+healthcare+professionals&as_epq=cancer+genomics&as_oq=&as_eq=&as_nlo=&as_nhi=&lr=&cr=&as_qdr=all&as_sitesearch=&as_occt=body&safe=images&as_filetype=&as_rights=  

and b) in Bing:           https://www.bing.com/search?q=cancer+genomics%2c+education+initiatives%2c+training+and+course+for+healthcare+professionals&filters=ex1%3a%22ez5_12053_18261%22&mkt=it-it&httpsmsn=1&msnews=1&refig=b2739f8c60eb4a2fbb6ef6ed8cce8f5d&sp=-1&pq=cancer+genomics%2c+education+initiatives%2c+course+for+healthcare+professionals&sc=0-75&qs=n&cvid=b2739f8c60eb4a2fbb6ef6ed8cce8f5d&qpvt=cancer+genomics%2c+education+initiatives%2c+training+and+course+for+healthcare+professionals

Following the reviewer suggestion, we conducted the search using the same search terms, also in “DuckDuckgo”, including only items published until October 2021, in order to have the same search period for all the browsers (https://duckduckgo.com/?q=cancer+genomics+education+initiatives+training+and+course+for+healthcare+professionals+%22cancer+genomics%22&t=h_&ia=web)

After the initial search, we retrieved 149 items that were carefully explored by title and description, and afterwards a total of three were included. We updated the flowchart, tables and the results section adding the new data.

Comment 4: Lines 302-310: the authors claim that only three countries (USA, UK and Japan) incorporated genomics into student classes as of 2018. This is highly incorrect, as, alredy in 2018, there were several university classes on genomics across Europe, and even entire BSc/MSc courses named "Genomics" (in Leuven, Paris, Bologna, Dublin, Milano, Nijmegen...)

Answer: We thank the reviewer for this valuable comment. It has not been reported clearly previously in the manuscript that the survey reported the training directed to nurses, and according to the data of 18 countries explored by the survey, only the mentioned countries (Japan, UK and USA) at the time of the study had incorporated genomics into students classes for nursing education. We apologize for the misunderstanding, we corrected the text and also we added at the discussion section (second paragraph, page 22), the data from a systematic review by Talwae et al. 2019, that summarized the existing genomics programs and evaluation for health professional students that reported that the majority of curricula were not theory based and were offered to medical and pharmacy students.

Comment 5: Line 303: typo "continously" should be "continuous"

Answer: We thank the reviewer for the warning. We have revised the sentence according to his/her suggestion.

Reviewer 2 Report

Authors can includes few examples of "why there is a need of personalized medicine based on genomics" and how it has proved to be beneficial. 

Author Response

Dear Editor, Dear Reviewer,

We would like to thank you for your valuable comments and for the opportunity to resubmit our work. We have amended the paper according to the received suggestions and we hope that it now appears improved.

Hereafter point-to-point answers are provided.

Best regards

Reviewer 2

Authors can includes few examples of "why there is a need of personalized medicine based on genomics" and how it has proved to be beneficial. 

Answer: We thank the reviewer for the valid suggestions; we added the examples that helped to improve our paper at the discussion section, second paragraph:  “Through a personalized medicine based on genomics has been improved risk stratification, for example in hereditary breast cancer. Carriers of BRCA1/2 pathogenic variant carriers and their relatives have an increased risk of breast cancer due to their genetic predisposition, and are a well-established sub-group of individuals with needs for targeted prevention and care pathways based on a risk-based approach Furthermore, researches are using whole exome analysis to characterize the genomic landscape of responders and non/responders to anti-HER2 treatment among metastatic breast cancer patients. Genomics is contributing to understand the polygenic nature of certain disease, such as cardiovascular disease, or diabetes.  Polygenic score models are being developed for specific subgroups of cardiovascular disease, such as CAD, stroke, and hypertension, that are currently used for screening and risk assessment to provide inform decisions for targeted treatments or tailored lifestyle modifications. Polygenic scores might be integrated into national programs to improve the predictive accuracy of cardiovascular risk assessment at population level.”

Round 2

Reviewer 1 Report

I am very satisfied by the effort the authors put in improving the manuscript, and I approve of it in the present form.